# Association between Long Working Hours and Psychological Distress: The Effect Modification by Request to Stay Home When Sick in the Workplace during the COVID-19 Pandemic

**DOI:** 10.3390/ijerph19073907

**Published:** 2022-03-25

**Authors:** Ayako Hino, Akiomi Inoue, Kosuke Mafune, Mayumi Tsuji, Seiichiro Tateishi, Akira Ogami, Tomohisa Nagata, Keiji Muramatsu, Yoshihisa Fujino

**Affiliations:** 1Department of Mental Health, Institute of Industrial Ecological Sciences, University of Occupational and Environmental Health, 1-1 Iseigaoka, Yahatanishi-ku, Kitakyushu 807-8555, Japan; kmafune@med.uoeh-u.ac.jp; 2Institutional Research Center, University of Occupational and Environmental Health, 1-1 Iseigaoka, Yahatanishi-ku, Kitakyushu 807-8555, Japan; akiomi@med.uoeh-u.ac.jp; 3Department of Environmental Health, School of Medicine, University of Occupational and Environmental Health, 1-1 Iseigaoka, Yahatanishi-ku, Kitakyushu 807-8555, Japan; tsuji@med.uoeh-u.ac.jp; 4Disaster Occupational Health Center, Institute of Industrial Ecological Sciences, University of Occupational and Environmental Health, 1-1 Iseigaoka, Yahatanishi-ku, Kitakyushu 807-8555, Japan; tateishi@med.uoeh-u.ac.jp; 5Department of Work Systems and Health, Institute of Industrial Ecological Sciences, University of Occupational and Environmental Health, 1-1 Iseigaoka, Yahatanishi-ku, Kitakyushu 807-8555, Japan; gamisan@med.uoeh-u.ac.jp; 6Department of Occupational Health Practice and Management, Institute of Industrial Ecological Sciences, University of Occupational and Environmental Health, 1-1 Iseigaoka, Yahatanishi-ku, Kitakyushu 807-8555, Japan; tomohisa@med.uoeh-u.ac.jp; 7Department of Preventive Medicine and Community Health, School of Medicine, University of Occupational and Environmental Health, 1-1 Iseigaoka, Yahatanishi-ku, Kitakyushu 807-8555, Japan; km@med.uoeh-u.ac.jp; 8Department of Environmental Epidemiology, Institute of Industrial Ecological Sciences, University of Occupational and Environmental Health, 1-1 Iseigaoka, Yahatanishi-ku, Kitakyushu 807-8555, Japan; zenq@med.uoeh-u.ac.jp

**Keywords:** long working hours, psychological distress, COVID-19, cross-sectional studies, Japan

## Abstract

The effect of workplace infection control measures required by the COVID-19 pandemic on the association between long working hours and psychological distress has not yet been fully revealed. This study investigated the effect of requesting to stay home when sick (RSH) on the association between long working hours and psychological distress. We conducted a cross-sectional survey in December 2020 among participants who had previously registered with a Japanese web survey company. A total of 27,036 workers completed a self-administered questionnaire which assessed usual daily overtime work hours. RSH was assessed using an original single-item scale, while psychological distress was measured with the K6 scale. After the interaction effect of overtime work hours and RSH on psychological distress was tested, we conducted stratified analyses using RSH. The statistical analysis demonstrated a significant interaction effect (*p* for interaction < 0.001). When we conducted stratified analyses, the odds ratios increased with longer working hours, both with and without RSH groups; however, the risk of long working hours causing psychological distress was greater in the latter group (odds ratio = 1.95 [95% confidence interval: 1.62–2.36] than in the former group (odds ratio = 1.73 [95% confidence interval: 1.55–1.93]). We found that working without RSH could strengthen the association between long working hours and psychological distress. Our findings contribute to preventing the deterioration of mental health during the COVID-19 pandemic.

## 1. Introduction

The coronavirus disease 2019 (COVID-19), a respiratory disease caused by SARS-CoV-2, has resulted in a global pandemic. In Japan, the first case of infection was reported in early 2020; the infection spread rapidly, thereafter. The first epidemic wave emerged in April 2020, the second in July 2020, and the third in December 2020; moreover, ongoing waves continue up until today. The Japanese government implemented community-based measures, similar to those applied globally, to control the pandemic, including quarantining, encouraging people’s prevention practices, restricting travel and events, and promoting social distancing [1]. However, these infection control measures should be implemented by not only individuals but also companies. Therefore, several workplace measures have been implemented to control COVID-19 infections worldwide.

Several previous studies in Japan have reported the association between workplace infection measures and mental health. A cross-sectional study reported that several workplace measures result in better psychological distress and job performance [2]. Another cross-sectional study revealed that companies with few measures had significantly worse mental health compared with those with sufficient measures [3]. These previous studies included multiple workplace infection measures such as preventive measures taken including those implemented by individuals (e.g., hand washing, wearing masks), preventive measures to reduce the risk of infection at the workplace (e.g., refrain from business trips, restricting outside visitors), and criteria and procedures for waiting at home and clinical contact (e.g., request to refrain from going to work when ill, report request for fever). Among others, Sasaki et al. [2] reported that “criteria and procedures for waiting at home and clinical contact” significantly increased job performance. Another study reported a similar tendency, which showed that “requesting employees not come to work when they are not feeling well” was highly associated with decreased psychological distress among workplace infection control measures [3]. Requesting to stay home when sick (i.e., criteria for deciding whether or not employees should come to work when they are unwell: hereinafter referred to as RSH) may be particularly important for mental health as compared with other infection control measures.

On the other hand, long working hours are considered a problem in workplaces worldwide. The association between long working hours and physical health has been reported in some previous studies [4,5]. Moreover, similar evidence was reported for mental health [6]. While numerous studies during the COVID-19 pandemic have revealed the association between long working hours and mental health, the majority of such studies have been conducted on healthcare workers. For example, a cross-sectional study in a Japanese healthcare institution showed that increasing working hours might be a risk factor for depression among nurses [7]. Further, a prior cross-sectional study conducted in foreign countries (i.e., not including Japan) found that job strain, including increased working hours, was the most significant psychosocial stressor among German healthcare workers [8]. However, during the COVID-19 pandemic, along with healthcare workers, other general workers may also be exposed to overworking. The International Labour Organization (ILO) guidelines [9] lists long working hours as an important psychosocial risk during the COVID-19 pandemic, indicating that the psychological effects of long working hours are an important issue for essential workers other than healthcare workers.

Furthermore, during the COVID-19 pandemic, long working hours may increase the possibility of infection because of longer exposure to infection sources in the workplace. As a result, there would be a strong association between long working hours and psychological distress in workplaces with inadequate infection control measures. However, the effect of workplace infection control measures on the association between long working hours and psychological distress has not yet been revealed. In this study, we focused particularly on RSH among workplace infection control measures and investigated the effect of RSH on the association between long working hours and psychological distress. This study aimed to reveal the effect modification by RSH on the association between long working hours and psychological distress. We hypothesized that the association between long working hours and psychological distress would be stronger for workers in workplaces without RSH than those with RSH.

## 2. Materials and Methods

### 2.1. Participants

We conducted a cross-sectional survey from 22–26 December 2020, among participants who had previously registered with a Japanese web survey company. At the time of this survey, the third wave of the COVID-19 pandemic had occurred in Japan, and the number of infected persons was increasing rapidly. Details of the protocol for this study have been previously reported [10]. The participants were selected by allocating an equal distribution of numbers by gender, occupation, and residential region, and 33,302 were eligible to respond. Participants with exceptionally short response times (≤6 min), exceptionally low body weight (<30 kg), exceptionally short height (<140 cm), and those who provided inconsistent responses to comparable questions (e.g., inconsistent responses to questions on marital status and area of residence), as well as incorrect responses to a question (i.e., those who gave fraudulent responses; select the third largest number from among the following five numbers) were excluded. After excluding these invalid responses, the total number of respondents included in the final analysis was 27,036.

The purpose and procedures of the study were explained to the participants, and informed consent was obtained through an online website. The Ethics Committee of Medical Research, University of Occupational and Environmental Health, Japan reviewed and approved the study procedures (approval number: R2-079).

### 2.2. Measures

#### 2.2.1. Overtime Work Hours

We obtained information on usual daily overtime work hours using a self-administered questionnaire. We explored overtime using the question, “How many hours of overtime do you work per day?”. Following the classification standard used in a previous systematic review of overtime work hours [11], overtime work hours per day were classified into three groups: almost none, <2 h, and ≥2 h, which is equivalent to less than 40 h, <50 h, and ≥50 h of work per week, including regular working hours, respectively.

#### 2.2.2. Request to Stay Home When Sick (RSH)

A single-item scale was used to investigate RSH using the question, “Does your workplace request employees to refrain from coming in to work when they are not feeling well?” in the section of the questionnaire asking about infection control measures. We defined those responding with “yes” as “with RSH” and those answering “no” as “without RSH”.

#### 2.2.3. Psychological Distress

In this survey, psychological distress was measured using the K6 scale [12,13]. This scale has been widely used in Japan as well as in foreign countries to measure the symptoms of psychological distress in the past 30 days. The K6 scale comprises a six-item scale measuring the extent of psychological distress in the past 30 days with a five-point response option ranging from 0 = none of the time to 4 = all of the time (response range, 0–24). In this sample, Cronbach’s α coefficient was 0.93. Participants were dichotomized into those with psychological distress (total K6 score of five or more) and those without (0–4 score) using a cutoff point recommended for the Japanese population [14].

#### 2.2.4. Other Covariates

Other covariates included demographic characteristics (i.e., gender, age, marital status, residence status, educational background, present illness, and residence area) and occupational characteristics (i.e., job type, frequency of remote working, and psychosocial work characteristics). Age (response range 20–65 years old) was treated as a continuous variable. Marital status was classified into three groups: married, divorced or widowed, and unmarried. Residence status was classified into two groups: living with family and living alone. Educational background was classified into six groups: junior high school, high school, vocational school, junior college or technical college, university, and graduate school. Present illness was classified into two groups: none and any. Residence areas were classified into two groups: high infection rate areas (i.e., prefectures with a declaration of a state of emergency by the Japanese Government due to COVID-19 [15]) and low infection rate areas (i.e., prefectures without a declaration of a state of emergency by the Japanese Government due to COVID-19). Job type was classified into three groups: mainly desk work (e.g., clerical, computer operations), jobs mainly involving interpersonal communication (e.g., customer service, sales), and mainly labor (e.g., production work, physical work, nursing care). The frequency of remote working was classified into five groups: ≥4 times/week, ≥2 times/week, ≥1 time/week, ≥1 time/month, and almost none. Psychosocial work characteristics, including job control, supervisor support, and coworker support, were assessed using the Japanese version of the Job Content Questionnaire (JCQ) [16,17]. The JCQ includes a nine-item job control scale (response range 24–96, Cronbach’s alpha in the present sample = 0.74), a four-item coworker support scale (response range 4–16, Cronbach’s alpha in the present sample = 0.94), and a four-item coworker support scale (response range 4–16, Cronbach’s alpha in the present sample = 0.90).

### 2.3. Statistical Analysis

We conducted multiple logistic regression analyses to estimate the prevalence of odds ratios (ORs) and their 95% confidence intervals (CIs) of psychological distress (defined as having a score of five or more on the K6 scale). In the analyses, we first tested the main effects of overtime work hours and RSH on psychological distress. Thereafter, we tested the interaction effect of overtime work hours and RSH on psychological distress. When we observed a significant interaction effect, we conducted stratified analyses using RSH. In a series of analyses, we first calculated the crude ORs (i.e., without any adjustment) (Model 1). Then, we adjusted for demographic characteristics (i.e., gender, age, marital status, residence status, educational background, present illness, and residence area) (Model 2) and additionally adjusted for occupational characteristics (i.e., job type, frequency of remote working, and psychosocial work characteristics) (Model 3). The level of significance was set at *p* < 0.05. Statistical analyses were performed using IBM Statistical Package for the Social Sciences (SPSS) Statistics version 22 (SPSS Inc., Chicago, IL, USA).

## 3. Results

Participants’ demographic and occupational characteristics are presented in Table 1. Of all participants, 20,230 (74.8%) worked with RSH and 6806 (25.2%) worked without RSH. Compared with the participants with RSH, those without RSH were more likely to have no overtime work and had higher K6 scores. Furthermore, participants without RSH were more likely to be male, unmarried, living alone, mainly engaged in labor, and have a lower educational background than those with RSH.

Table 2 shows the main effect of overtime work hours on psychological distress. The <2 h group (ORs = 1.41–1.43; 95% CIs: 1.34–1.51) and ≥2 h group (ORs = 1.69–1.81; 95% CIs: 1.55–1.99) have significantly higher prevalence ORs of psychological distress compared to the “almost no overtime” group in any models. For RSH, the “without RSH” group has significantly higher prevalence ORs of psychological distress than the “with RSH” group in any models (ORs = 1.34–1.54; 95% CIs: 1.18–1.64). Focusing on the other covariates, for present illness, the “any” group has significantly higher prevalence ORs of psychological distress compared to the “none” group in any models (ORs = 1.99–2.01; 95% CIs: 1.89–2.13). On the other hand, for coworker support, the higher the score, the significantly lower the OR with psychological distress (OR = 0.92; 95% CI: 0.91–0.93). The interaction effects of overtime work hours and RSH on psychological distress were significant in any models (*p* for interaction < 0.001).

When we conducted stratified analyses by RSH, in the “with RSH” group, the <2 h subgroup (ORs = 1.31–1.32; 95% CIs: 1.24–1.41) and ≥2 h subgroup (ORs = 1.60–1.74; 95% CIs: 1.45–1.94) had significantly higher ORs of psychological distress compared with the “almost no overtime” subgroup in all models (Table 3). A similar tendency was observed in the “without RSH” group; however, the ORs were greater in the “without RSH” group than the “with RSH” group.

## 4. Discussion

This study demonstrated the significant main effects and interaction effect of overtime work hours and RSH on psychological distress. For the other covariates, present illness was significantly positively associated with psychological distress, while coworker support was significantly negatively associated with psychological distress.

Long working hours were found to be significantly associated with deteriorating mental health in this study. The present results are consistent with those of previous studies conducted before the COVID-19 pandemic [11] and previous studies conducted among healthcare workers during the COVID-19 pandemic [7,8]. The findings indicate that taking measures to avoid long working hours is useful for preventing mental health deterioration not only during normal times but also during the COVID-19 pandemic. Moreover, an absence of RSH was a significant risk factor of psychological distress. Previous studies have described RSH as a particularly important risk factor of deteriorating mental health among workplace infection control measures [2,3], a finding that was replicated in the present study. Therefore, both long working hours and an absence of RSH were revealed to be risk factors of mental health deterioration.

This study demonstrated a significant interaction effect of overtime work hours and RSH on psychological distress. Workers without RSH had a significantly higher risk of psychological distress due to long working hours than those who worked with RSH. As mentioned earlier, previous studies reported that long working hours [7,8] and RSH [2,3] are important for mental health. Our study extends prior findings by revealing that the absence of RSH in the workplace could result in a stronger association between long working hours and psychological distress. Workers working long hours experience a poor quality and quantity of sleep, fatigue, and disruption of family and social activities, which results in poor mental health [18]. Long working hours, during a pandemic, may have an additional impact on poor mental health due to the increasing possibility of infection because of the prolonged time spent in the workplace. During the influenza A (H1N1) pdm09 outbreak in 2009, a workplace policy to allow sick employees to take leave was associated with a lower likelihood of overall workplace infections [19,20]. Thus, working with RSH during a pandemic may have had a better impact on mental health by decreasing the anxiety of contracting an infection at work. Therefore, in the absence of RSH, the association between long working hours and mental health may become stronger during the COVID-19 pandemic.

However, it should be noted that setting RSH does not completely buffer psychological distress owing to long working hours. The overtime group (i.e., <2 h and ≥2 h subgroups) had higher ORs, compared with the group with no overtime, even among workers with RSH. Thus, workplaces should not only set RSH but also promote the usual measures of reduction of long working hours during the COVID-19 pandemic.

Having any present illness was a significant risk factor of psychological distress. These illnesses included both mental and physical health problems. It was suggested that having any present illness, not limited to mental health disorders, may deteriorate psychological distress. On the other hand, higher levels of coworker support significantly decreased psychological distress. Previous studies conducted during the COVID-19 pandemic also reported that a lack of coworker support increased mental health problems [21], which is consistent with our results. Coworker support has been reported to be effective in reducing tension and problem solving by allowing workers to share their emotions with each other [22]. During the COVID-19 pandemic, having a familiar advisor, such as a coworker, may be effective in preventing workers’ mental health deterioration.

Some limitations of this study should be considered. First, this was a cross-sectional study; therefore, causality is unclear. Moreover, those with mental health issues may work longer hours because of reduced work efficiency. However, in general, in companies in Japan, occupational health staff are assigned to workplaces to ensure that those with mental health issues usually do not work long hours. Therefore, while it is unlikely that causality will be reversed, future longitudinal studies are needed. Second, we assessed overtime work hours using self-administered questionnaires, which may have resulted in a bias. However, a previous study revealed that the correlation between self-reported and company records of working hours was quite high [23]; thus, the effect may not be large. Third, to assess overtime work hours, we examined overtime work hours per day; however, previous studies used the average overtime work hours per month or per week [24,25]. Thus, the workload evaluation may not be accurate in the case of occupations with a greater change in daily working hours. Moreover, we adjusted for the job type, but we did not survey the details of the occupation. Fourth, we surveyed RSH with a single-item self-administered questionnaire. Participants with deteriorating mental health may have a biased perspective of the policies of the company (i.e., even if the company has RSH, they may have answered that the company has no RSH) which may lead to an overestimation of the present results. Future studies should collect objective information regarding RSH to check whether the present results can be replicated. Fifth, we did not survey “whether the workplace compensates sick pay during absence.” Since sick pay allows employees to take time off from work without worry, a lack of sick pay could worsen mental health [8]. As the RSH measured in this study may include those without sick pay during absence, the amplifying effect of an absence of RSH found in this study may be underestimated. Thus, future studies should not only investigate RSH but also sick pay for absence from work. Sixth, although the interaction effect of overtime work hours and RSH on psychological distress was statistically significant, the dose–response relationship between overtime work hours and psychological distress was observed regardless of RSH, which pointed to an increased risk of psychological distress with longer overtime work hours. Therefore, whether the observed interaction effect is clinically significant or not needs further research. Seventh, we classified job type into three groups: mainly desk work, jobs mainly involving interpersonal communication, and labor. However, this classification of job type was not precise, and it may have caused a misclassification of job type categorization. The questionnaire includes examples of each job type (e.g., clerical, customer service, production work), and the participants should have selected one that is as close as possible to their own job type; nonetheless, it would be desirable to use a more precise classification in the future. Finally, there was a lack of information regarding the possible confounding factors affecting mental health. Workplaces with RSH are likely to have better workplace factors (e.g., good benefits and good job descriptions). Although we adjusted for job type, frequency of remote working, and psychosocial work characteristics, the effects may not be fully eliminated. Future research should also investigate additional workplace factors to eliminate their influence.

## 5. Conclusions

In conclusion, this study revealed an amplifying effect of the absence of RSH on the association between long working hours and psychological distress. We revealed that working without RSH could strengthen the association between long working hours and mental health during the COVID-19 pandemic. The ILO guideline [9] and guidelines based on the National Institute for Occupational Safety and Health (NIOSH) Total Worker Health (TWH) program [26] highlight the importance of informing workers about sick leave policies, in addition to preventing long working hours. Our findings support the ILO and NIOSH TWH guidelines and are useful for preventing the deterioration of mental health during a pandemic.

## Figures and Tables

**Table 1 ijerph-19-03907-t001:** Participants’ demographic and occupational characteristics, overtime work hours, and psychological distress by RSH † (*n* = 27,036).

	With RSH		Without RSH		
	(*n* = 20,230)		(*n* = 6806)		
	Mean (SD)	*n* (%)	Mean (SD)	*n* (%)	Cronbach’s α
Gender					
Men		10,025 (49.6)		3789 (55.7)	
Women		10,205 (50.4)		3017 (44.3)	
Age	46.7 (10.6)		47.9 (10.3)		
Marital status					
Married		11,625 (57.5)		3406 (50.0)	
Divorce, widowed		2035 (10.1)		808 (11.9)	
Unmarried		6572 (32.5)		2592 (38.1)	
Residence status					
Living with family		15,989 (79.0)		5240 (77.0)	
Living alone		4241 (21.0)		1566 (23.0)	
Educational background					
Junior high school		214 (1.1)		154 (2.3)	
High school		4805 (23.8)		2148 (31.6)	
Vocational school		2643 (13.1)		1030 (15.1)	
Junior college, technical college		2188 (10.8)		683 (10.0)	
University		9187 (45.4)		2534 (37.2)	
Graduate school		1193 (5.9)		257 (3.8)	
Present illness					
None		13,057 (64.5)		4469 (65.7)	
Any		7173 (35.5)		2337 (34.3)	
Residence area					
Low infection rate areas		11,855 (58.6)		3979 (58.5)	
High infection rate areas		8375 (41.4)		2827 (41.5)	
Job type					
Mainly desk work		10,303 (50.9)		3165 (46.5)	
Jobs mainly involving interpersonal communication		5267 (26.0)		1660 (24.4)	
Mainly labor		4660 (23.0)		1981 (29.1)	
Frequency of remote working					
≥4 times/week		1702 (8.4)		1088 (16.0)	
≥2 times/week		1236 (6.1)		241 (3.5)	
≥1 time/week		745 (3.7)		133 (2.0)	
≥1 time/month		530 (2.6)		85 (1.2)	
Almost none		16,017 (79.2)		5259 (77.3)	
Psychosocial work characteristics (JCQ) ‡					
Job Control	63.5 (11.3)		63.0 (10.3)		0.74
Supervisor support	10.4 (2.8)		9.0 (3.3)		0.94
Coworker support	10.8 (2.4)		9.7 (3.0)		0.90
Overtime work hours (daily average)					
Almost none		8945 (44.2)		3898 (57.3)	
<2 h		9417 (46.5)		2323 (34.1)	
≥2 h		1868 (9.2)		585 (8.6)	
Psychological distress (K6)	4.4 (5.2)		5.5 (6.0)		0.93
Number of cases §		7678 (38.0)		3139 (46.1)	

† RSH, request to stay home when sick. ‡ JCQ, Job Content Questionnaire. § Psychological distress was defined as having a score of five or more on the K6 scale.

**Table 2 ijerph-19-03907-t002:** Main effects of overtime work hours, RSH †, and other covariates on psychological distress ‡: multiple logistic regression analysis.

Main Effect			Odds Ratio (95% Confidence Interval)		
	*n*	No. of Cases (%)	Model 1	Model 2 §	Model 3 ||
Overtime work hours					
Almost none	12,843	4585 (35.7)	1.00	1.00	1.00
<2 h	11,740	5062 (43.1)	1.43 (1.35–1.50)	1.41 (1.34–1.49)	1.43 (1.35–1.51)
≥2 h	2353	1170 (47.7)	1.69 (1.55–1.85)	1.81 (1.65–1.98)	1.81 (1.65–1.99)
Request to stay home when sick (RSH)					
With RSH	20,230	7678 (38.0)	1.00	1.00	1.00
Without RSH	6806	3139 (46.1)	1.47 (1.39–1.56)	1.54 (1.46–1.64)	1.34 (1.18–1.33)
Gender					
Men	13,814	4779 (34.6)		1.00	1.00
Women	13,222	6038 (45.7)		1.21 (1.14–1.29)	1.25 (1.18–1.33)
Age ¶				0.97 (0.96–0.97)	0.97 (0.96–0.97)
Marital status					
Married	15,029	4585 (35.7)		1.00	1.00
Divorce, widowed	2843	5062 (43.1)		1.38 (1.26–1.51)	1.32 (1.21–1.45)
Unmarried	9164	1170 (47.7)		1.29 (1.21–1.38)	1.23 (1.15–1.31)
Residence status					
Living with family	21,229	8161 (38.4)		1.00	1.00
Living alone	5807	2656 (45.7)		1.05 (0.98–1.13)	1.04 (0.97–1.12)
Educational background					
Junior high school	368	179 (48.6)		1.00	1.00
High school	6953	2787 (40.1)		0.75 (0.61–0.94)	0.75 (0.60–0.94)
Vocational school	3673	1599 (43.5)		0.78 (0.63–0.98)	0.81 (0.65–1.02)
Junior college, technical college	2871	1226 (42.7)		0.77 (0.62–0.97)	0.80 (0.63–1.01)
University	11,721	4498 (38.4)		0.69 (0.56–0.86)	0.74 (0.59–0.92)
Graduate school	1450	528 (36.4)		0.64 (0.50–0.81)	0.72 (0.56–0.92)
Present illness					
None	17,526	6304 (36.0)		1.00	1.00
Any	9510	4513 (47.5)		1.99 (1.89–2.11)	2.01 (1.90–2.13)
Residence area					
Low infection rate areas	15,834	6465 (40.8)		1.00	1.00
High infection rate areas	11,202	4352 (38.9)		0.96 (0.91–1.01)	0.94 (0.89–0.99)
Job type					
Mainly desk work	13,468	5217 (38.7)			1.00
Jobs mainly involving interpersonal communication	6927	2791 (40.3)			1.04 (0.98–1.11)
Mainly labor	6641	2809 (42.3)			1.10 (1.03–1.17)
Frequency of remote working					
≥4 times/week	2790	1077 (38.6)			1.00
≥2 times/week	1477	567 (38.4)			1.04 (0.91–1.20)
≥1 time/week	878	339 (38.6)			1.03 (0.87–1.21)
≥1 time/month	615	255 (41.5)			1.17 (0.97–1.41)
Almost none	21,276	8579 (40.3)			0.90 (0.82–0.99)
Psychosocial work characteristics (JCQ) ††					
Job Control					0.99 (0.99–0.99)
Supervisor support					0.96 (0.95–0.97)
Coworker support					0.92 (0.91–0.93)

† RSH, request to stay home when sick. ‡ Psychological distress was defined as having a score of five or more on the K6 scale. § Adjusted for demographic characteristics (i.e., gender, age, marital status, residence status, educational background, present illness, and residence area). || Additionally adjusted for occupational characteristics (i.e., job type, frequency of remote working, and psychosocial work characteristics). ¶ Treated as continuous variables. †† JCQ, Job Content Questionnaire. Treated as continuous variables.

**Table 3 ijerph-19-03907-t003:** Association between overtime work hours and psychological distress † by RSH ‡: multiple logistic regression analysis.

Overtime Work Hours			Odds Ratio (95% Confidence Interval)		
	*n*	No. of Cases (%)	Model 1 §	Model 2 ||	Model 3 ¶
With RSH					
Almost none	8945	3038 (34.0)	1.00	1.00	1.00
<2 h	9417	3796 (40.3)	1.31 (1.24–1.39)	1.32 (1.24–1.40)	1.32 (1.24–1.41)
≥2 h	1868	844 (45.2)	1.60 (1.45–1.77)	1.74 (1.56–1.94)	1.73 (1.55–1.93)
*p* for linear trend			<0.001	<0.001	<0.001
Without RSH					
Almost none	3898	1547 (39.7)	1.00	1.00	1.00
<2 h	2323	1266 (54.5)	1.82 (1.64–2.02)	1.75 (1.57–1.95)	1.77 (1.58–1.98)
≥2 h	585	326 (55.7)	1.91 (1.61–2.28)	1.95 (1.63–2.35)	1.95 (1.62–2.36)
*p* for linear trend			<0.001	<0.001	<0.001

† Psychological distress was defined as having a score of five or more on the K6 scale. ‡ RSH, request to stay home when sick. § Crude (i.e., without any adjustment). || Adjusted for demographic characteristics (i.e., gender, age, marital status, residence status, educational background, present illness, and residence area). ¶ Additionally adjusted for occupational characteristics (i.e., job type, frequency of remote working, and psychosocial work characteristics).

## Data Availability

The data presented in this study are not available to the public due to ethical restrictions.

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
