# Peer review of "Association between Long Working Hours and Psychological Distress: The Effect Modification by Request to Stay Home When Sick in the Workplace during the COVID-19 Pandemic"

_ijerph, 2022, doi:10.3390/ijerph19073907_

Round 1

Reviewer 1 Report

This manuscript presents an interesting topic relating to work hours, psychological distress, and sick leave criteria. This paper was well written for the most part, but the introduction and aims should be clarified prior to publication. Well done.

  • The title is a little unclear – reading it I am not sure how sick leave criteria relates to work hours/psychological distress. Suggest rewording.

Abstract

  • It is unclear whether participants completed one survey or many (ln 33“questionnaires”).
  • No background information is included in the abstract – I am not sure why this study is necessary, or in fact what is being assessed.
  • Your findings and conclusions are not clear because the question being asked has not been provided. E.g., ln 39 “the risk was greater” – the risk of what?

Introduction

  • I’m not sure if the authors mean that workplace control measures e.g., not going to work when feeling sick improve psychological distress/job performance generally, or as compared with other pandemic-related control measures (paragraph 1, pg 2)
  • The link between karoshi/karojisatsu and your topic is unclear
  • Can you please explain why the pandemic may increase working hours for non-healthcare workers?
  • Regarding the link between long working hours à infection risk à psychological distress. Do you mean that an increased RISK of infection results in greater psychological distress, or that actually getting infected results in psychological distress (or both?). Ln 94
  • I’m not sure what you mean by sick leave criteria and how this relates to your topic.
  • What is your research question? This should be explained more clearly.

Methods

  • Looking at table 1, I do not understand what you mean by with/without sick leave criteria. This should be explained in the text when Table 1 is referred to. I would move table 1 to the results section.
  • I would argue that desk work and interpersonal communication work could be the same job. What is the difference? How was this determined? What if, for example, the individual worked in a call center or similar – presumably this would be both.
  • Ln 132 – what if the individual did inconsistent overtime? E.g., 5h on one day, 0h on another? Do you mean >50h of overtime (i.e., in addition to regular work hours)?? That seems like a lot!
  • I am still not sure what you mean by ‘sick leave criteria’. This seems to be referring to whether they have sick leave available at all?

Results

  • Did you ask about the type of contract participants were on? I imagine the individuals with no sick leave would most likely be on casual contracts (i.e., unstable working arrangements) and may therefore be more distressed as a result of that – rather than sick leave alone.
  • The results section is very short. It may be helpful to explore some other factors likely to impact distress (e.g., age, gender, work hours, job control, infection rates in their area, etc). This should also be addressed in the discussion (i.e., potential confounders)

Discussion

  • Surely the amount of overtime would play a large role in distress (i.e., 2h vs10h vs 40h)

Author Response

First, we appreciate your positive evaluation of the present study and your useful comments regarding our manuscript. According to your comments, we have carefully revised our manuscript considering the following points.

Reviewer #1:

This manuscript presents an interesting topic relating to work hours, psychological distress, and sick leave criteria. This paper was well written for the most part, but the introduction and aims should be clarified prior to publication. Well done.

The title is a little unclear – reading it I am not sure how sick leave criteria relates to work hours/psychological distress. Suggest rewording.

<Response>

According to your suggestion, we change the title “The effect of sick leave criteria” to “The effect modification by request to stay home when sick” to convey the content of the manuscript.

Abstract

It is unclear whether participants completed one survey or many (ln 33“questionnaires”).

<Response>

We apologize for the misleading wording; I have corrected it to “a self-administered questionnaire” so that it is clear that it was one survey (page 1, line 35).

No background information is included in the abstract – I am not sure why this study is necessary, or in fact what is being assessed.

<Response>

According to your suggestion, at the beginning of the abstract, the purpose for which this study was conducted has been added (page 1, lines 30–32).

Your findings and conclusions are not clear because the question being asked has not been provided. E.g., ln 39 “the risk was greater” – the risk of what?

<Response>

According to your suggestion, we specified “the risk of long working hours to cause psychological distress” (page 1, line 39–43).

Introduction

I’m not sure if the authors mean that workplace control measures e.g., not going to work when feeling sick improve psychological distress/job performance generally, or as compared with other pandemic-related control measures (paragraph 1, pg 2)

<Response>

As you pointed out, this meant that the workplace control measures, e.g., not going to work when feeling sick, improve psychological distress/job performance as compared with other pandemic-related control measures. We have revised the relevant section to make this clearer (page 2, lines 74–77).

The link between karoshi/karojisatsu and your topic is unclear

<Response>

As you pointed out, the link between karoshi/karojisatsu and our topic is unclear. We removed the relevant section and consolidated it into a review of previous studies on the relationship between working hours and mental health (page 2, lines 78–81).

Can you please explain why the pandemic may increase working hours for non-healthcare workers?

<Response>

We apologize for the lack of explanation. The term “non-healthcare workers” correctly refers to essential workers other than healthcare workers. We have corrected the relevant section to make this clear (page 2, lines 89–92).

Regarding the link between long working hours an infection risk a psychological distress.

Do you mean that an increased RISK of infection results in greater psychological distress, or that actually getting infected results in psychological distress (or both?). Ln 94

<Response>

We meant that an increased RISK of infection results in greater psychological distress.  We have revised the relevant section to make this clear (page 2, lines 93–96).

I’m not sure what you mean by sick leave criteria and how this relates to your topic.

<Response>

As you and Reviewer 4 pointed out, we changed “sick leave criteria” to “request to stay home when sick (RSH).” We have revised the relevant section to make it clear that RSH is one of the infection control measures (page 2, lines 98–100 and page 3, lines 136–138).

What is your research question? This should be explained more clearly.

<Response>

As you and Reviewer 3 pointed out, we revised the text to clarify the purpose and hypothesis of this study (page 2, line 100 to page 3, lines 104).

Methods

Looking at table 1, I do not understand what you mean by with/without sick leave criteria. This should be explained in the text when Table 1 is referred to. I would move table 1 to the results section.

<Response>

According to your suggestion, we moved Table 1 to the Results section (page 4, line 191).

I would argue that desk work and interpersonal communication work could be the same job. What is the difference? How was this determined? What if, for example, the individual worked in a call center or similar – presumably this would be both.

<Response>

The questionnaire includes examples of each job type, and the participants should have selected one that is as close as possible to their own job type. The examples are listed in the Materials and Methods section (page 4, lines 165–168). However, the classification of job type was not precise and it may have caused misclassification of job type categorization. We noted the possibility of misclassification as a limitation in the Discussion section (page 10, lines 323–329).

Ln 132 – what if the individual did inconsistent overtime? E.g., 5h on one day, 0h on another? Do you mean >50h of overtime (i.e., in addition to regular work hours)?? That seems like a lot!

<Response>

We apologize for the lack of explanation. ≥50 hours per week indicated working hours that included regular working hours, not in addition to regular working hours. We have corrected the relevant section to make this clear (page 3, lines 129–133).

I am still not sure what you mean by ‘sick leave criteria’. This seems to be referring to whether they have sick leave available at all?

<Response>

As you and Reviewer 4 pointed out, we changed “sick leave criteria” to “request to stay home when sick (RSH).” We have revised the relevant section to make it clear that RSH is one of the infection control measures (page 2, lines 98–100 and page 3, lines 136–138).

Results

Did you ask about the type of contract participants were on? I imagine the individuals with no sick leave would most likely be on casual contracts (i.e., unstable working arrangements) and may therefore be more distressed as a result of that – rather than sick leave alone.

<Response>

As you pointed out, we re-analyzed the data, adjusting for the type of contract. Additionally adjusted for the type of contract, the interaction effect was significant (p for interaction < 0.001). When we conducted stratified analyses by RSH, in the “with RSH” group, the < 2 hours (OR = 1.33; 95 % CI: 1.24–1.42), and ≥ 2 hours subgroups (OR = 1.74; 95 % CI: 1.56–1.94) had significantly higher ORs of psychological distress compared with the almost no overtime subgroup. In the “without RSH” group, the < 2 hours subgroup (OR = 1.77; 95 % CI: 1.59–1.98), and ≥ 2 hours subgroup (OR = 1.97; 95 % CI: 1.63–2.37) had significantly higher ORs of psychological distress compared with the almost no overtime subgroup.

Additional adjustment for the type of contract did little to change the results compared with no additional adjustment. Therefore, the type of contract was not included in the analysis model in this study.

The results section is very short. It may be helpful to explore some other factors likely to impact distress (e.g., age, gender, work hours, job control, infection rates in their area, etc). This should also be addressed in the discussion (i.e., potential confounders)

<Response>

As you and Reviewer 3 pointed out, we added a test of the main effects of working hours, RSH, and other covariates on mental health. We have also added Results and Discussion section of the covariates that are particularly important risk and prevention factors, respectively (page 6, lines 204–213 and Table 2 and page 8, lines 247 to page 9 line 261 and page 9, line 284–293).

Discussion

Surely the amount of overtime would play a large role in distress (i.e., 2h vs10h vs 40h)

<Response>

Thank you for your supportive comments. We believe that a more accurate method of measuring working hours needs to be considered in the future.

We hope that the revised manuscript in now acceptable for publication in the International Journal of Environmental Research and Public Health. Thank you.

Reviewer 2 Report

In the paper (introduction, discussion) it is worth to pont out other health problems related to excessive work. 

Author Response

Reviewer #2:

In the paper (introduction, discussion) it is worth to point out other health problems related to excessive work.

<Response>

Thank you for your confirmation. We have completed the revised manuscript based on all reviewers' suggestions.

We hope that the revised manuscript in now acceptable for publication in the International Journal of Environmental Research and Public Health. Thank you.

Reviewer 3 Report

  • I see that the authors have put a hypothesis at the end of the introduction. However, I recommend that they also write it in the form of an aim.
  • A positive point of the manuscript is the large sample they have achieved, although they should specify their age range.
  • The methodology section is very correct, explaining everything the reader needs to know to understand the manuscript.
  • The results section seems to me to be too brief. For the large number of participants in the sample, I think the analysis is too short. They could have taken advantage and analysed more factors to obtain better results. I think that the results they present are too brief to be published in a quality journal.

In general, it seems to me a very interesting subject and one that needs to be explored in greater depth, but I believe that the results they present are very scarce. The study should be extended.

Author Response

First, we appreciate your positive evaluation of the present study and your useful comments regarding our manuscript. According to your comments, we have carefully revised our manuscript considering the following points.

Reviewer #3:

I see that the authors have put a hypothesis at the end of the introduction. However, I recommend that they also write it in the form of an aim.

<Response>

As you and Reviewer 1 pointed out, we revised the text to clarify the purpose and hypothesis of this study (page 2, line 100 to page 3, lines 104).

A positive point of the manuscript is the large sample they have achieved, although they should specify their age range.

<Response>

Thank you for your suggestion. We treated age as a continuous variable and noted that the answer range was 20–65 years old in the Materials and Methods section (page 4, lines 156–157).

The methodology section is very correct, explaining everything the reader needs to know to understand the manuscript.

<Response>

Thank you for your supportive comments.

The results section seems to me to be too brief. For the large number of participants in the sample, I think the analysis is too short. They could have taken advantage and analysed more factors to obtain better results. I think that the results they present are too brief to be published in a quality journal. In general, it seems to me a very interesting subject and one that needs to be explored in greater depth, but I believe that the results they present are very scarce. The study should be extended.

<Response>

As you and Reviewer 1 pointed out, we added a test of the main effects of working hours, RSH, and other covariates on mental health. We have also added Results and Discussion section of the covariates that are particularly important risk and prevention factors, respectively (page 6, lines 204–213 and Table 2 and page 8, lines 247 to page 9 line 161 and page 9, line 284–293).

We hope that the revised manuscript in now acceptable for publication in the International Journal of Environmental Research and Public Health. Thank you.

Reviewer 4 Report

Dear Authors, I had the occasion to read your manuscript and I found it really interesting,  well written and your conclusions clearly stated. Nevertheless, in my opinion there are some issues to be addressed to improve the readability and explicative impact of the paper, also considering its potential in the landscape of the studies related to the effects on OSH of the Covid19 pandemic.  

Some MAJOR and MINOR ISSUES I suggest should be addressed:

- The term “sickleave criteria” seems to be too broad respect to the question done “Does your workplace request employees to refrain from coming in to work when they are not feeling well?”.

Authors have reported this on limitation too, but I believe that the use of the term “Sickleave criteria” lead the readers to think to a more incisive set of procedures and/or measures put in place by the organizations than simply requesting employees to refrain from coming in to work when they are not feeling well. Nevertheless, this kind of question seems to include only the request to the employees to stay home, but to leave free to decide themselves. Studying long working hours, we must consider that there are several studies focussing on presenteeism, that refers to workers going work despite complaints and ill-health that prompt them to take rest and sick leave. I suggest changing the term with another one more as “request to stay home when sick” or “request to refrain from going to work when ill”.  

-References on the association between long working hours and health must be updated with the recent systematic reviews and meta-analysis on long working hours from the WHO/ILO Joint Estimates of the Work-related Burden of Disease and Injuries. And to this regard please also consider the different findings on long working hours and depression talking about the effect on mental health:

Alexis Descatha, Grace Sembajwe, Frank Pega, Yuka Ujita, Michael Baer, Fabio Boccuni, Cristina Di Tecco, Clement Duret, Bradley A. Evanoff, Diana Gagliardi, Lode Godderis, Seong-Kyu Kang, Beon Joon Kim, Jian Li, Linda L. Magnusson Hanson, Alessandro Marinaccio, Anna Ozguler, Daniela Pachito, John Pell, Fernando Pico, Matteo Ronchetti, Yves Roquelaure, Reiner Rugulies, Martijn Schouteden, Johannes Siegrist, Akizumi Tsutsumi, Sergio Iavicoli,

The effect of exposure to long working hours on stroke: A systematic review and meta-analysis from the WHO/ILO Joint Estimates of the Work-related Burden of Disease and Injury, Environment International, Volume 142, 2020, 105746, ISSN 0160-4120, https://doi.org/10.1016/j.envint.2020.105746

Jian Li, Frank Pega, Yuka Ujita, Chantal Brisson, Els Clays, Alexis Descatha, Marco M. Ferrario, Lode Godderis, Sergio Iavicoli, Paul A. Landsbergis, Maria-Inti Metzendorf, Rebecca L. Morgan, Daniela V. Pachito, Hynek Pikhart, Bernd Richter, Mattia Roncaioli, Reiner Rugulies, Peter L. Schnall, Grace Sembajwe, Xavier Trudel, Akizumi Tsutsumi, Tracey J. Woodruff, Johannes Siegrist, The effect of exposure to long working hours on ischaemic heart disease: A systematic review and meta-analysis from the WHO/ILO Joint Estimates of the Work-related Burden of Disease and Injury, Environment International, Volume 142, 2020,105739, ISSN 0160-4120, https://doi.org/10.1016/j.envint.2020.105739.

Reiner Rugulies, Kathrine Sørensen, Cristina Di Tecco, Michela Bonafede, Bruna M. Rondinone, Seoyeon Ahn, Emiko Ando, Jose Luis Ayuso-Mateos, Maria Cabello, Alexis Descatha, Nico Dragano, Quentin Durand-Moreau, Hisashi Eguchi, Junling Gao, Lode Godderis, Jaeyoung Kim, Jian Li, Ida E.H. Madsen, Daniela V. Pachito, Grace Sembajwe, Johannes Siegrist, Kanami Tsuno, Yuka Ujita, JianLi Wang, Amy Zadow, Sergio Iavicoli, Frank Pega, The effect of exposure to long working hours on depression: A systematic review and meta-analysis from the WHO/ILO Joint Estimates of the Work-related Burden of Disease and Injury, Environment International, Volume 155, 2021, 106629, ISSN 0160-4120, https://doi.org/10.1016/j.envint.2021.106629.

-Job type category are not completely clear. What “Mainly labor” includes?  What “interpersonal communication” refers to? I suggest to explain better in the measures.

- At page 7 paragraph 2, authors stated: “Thus, workplaces should not only set criteria but also promote the reduction of long working hours during the COVID-19 pandemic”.

I think that promote the reduction of long working hours is not just a matter of during the Covid-19 pandemic.

- Authors include working from home as control variable and this is very appreciate in consideration of the topic. Nevertheless, there are several recent studies showing that working from home is increasing hours of work and reducing the sick leave absences at the same time (workers tend to work sick from home..). Maybe you should investigate (even only reporting frequencies) if the more the hour of working at home, the higher the long working hours are in your sample.  

Author Response

First, we appreciate your positive evaluation of the present study and your useful comments regarding our manuscript. According to your comments, we have carefully revised our manuscript considering the following points.

Reviewer #4:

Dear Authors, I had the occasion to read your manuscript and I found it really interesting,  well written and your conclusions clearly stated. Nevertheless, in my opinion there are some issues to be addressed to improve the readability and explicative impact of the paper, also considering its potential in the landscape of the studies related to the effects on OSH of the Covid19 pandemic. 

Some MAJOR and MINOR ISSUES I suggest should be addressed:

- The term “sickleave criteria” seems to be too broad respect to the question done “Does your workplace request employees to refrain from coming in to work when they are not feeling well?”.

Authors have reported this on limitation too, but I believe that the use of the term “Sick leave criteria” lead the readers to think to a more incisive set of procedures and/or measures put in place by the organizations than simply requesting employees to refrain from coming in to work when they are not feeling well. Nevertheless, this kind of question seems to include only the request to the employees to stay home, but to leave free to decide themselves. Studying long working hours, we must consider that there are several studies focusing on presenteeism, that refers to workers going work despite complaints and ill-health that prompt them to take rest and sick leave. I suggest changing the term with another one more as “request to stay home when sick” or “request to refrain from going to work when ill”. 

<Response>

As you and Reviewer 1 pointed out, we changed “sick leave criteria” to “request to stay home when sick (RSH).” We have revised the relevant section to make it clear that RSH is one of the infection control measures (page 2, lines 98–100 and page 3, lines 136–138).

-References on the association between long working hours and health must be updated with the recent systematic reviews and meta-analysis on long working hours from the WHO/ILO Joint Estimates of the Work-related Burden of Disease and Injuries. And to this regard please also consider the different findings on long working hours and depression talking about the effect on mental health:

Alexis Descatha, Grace Sembajwe, Frank Pega, Yuka Ujita, Michael Baer, Fabio Boccuni, Cristina Di Tecco, Clement Duret, Bradley A. Evanoff, Diana Gagliardi, Lode Godderis, Seong-Kyu Kang, Beon Joon Kim, Jian Li, Linda L. Magnusson Hanson, Alessandro Marinaccio, Anna Ozguler, Daniela Pachito, John Pell, Fernando Pico, Matteo Ronchetti, Yves Roquelaure, Reiner Rugulies, Martijn Schouteden, Johannes Siegrist, Akizumi Tsutsumi, Sergio Iavicoli,

The effect of exposure to long working hours on stroke: A systematic review and meta-analysis from the WHO/ILO Joint Estimates of the Work-related Burden of Disease and Injury, Environment International, Volume 142, 2020, 105746, ISSN 0160-4120, https://doi.org/10.1016/j.envint.2020.105746

Jian Li, Frank Pega, Yuka Ujita, Chantal Brisson, Els Clays, Alexis Descatha, Marco M. Ferrario, Lode Godderis, Sergio Iavicoli, Paul A. Landsbergis, Maria-Inti Metzendorf, Rebecca L. Morgan, Daniela V. Pachito, Hynek Pikhart, Bernd Richter, Mattia Roncaioli, Reiner Rugulies, Peter L. Schnall, Grace Sembajwe, Xavier Trudel, Akizumi Tsutsumi, Tracey J. Woodruff, Johannes Siegrist, The effect of exposure to long working hours on ischaemic heart disease: A systematic review and meta-analysis from the WHO/ILO Joint Estimates of the Work-related Burden of Disease and Injury, Environment International, Volume 142, 2020,105739, ISSN 0160-4120, https://doi.org/10.1016/j.envint.2020.105739.

Reiner Rugulies, Kathrine Sørensen, Cristina Di Tecco, Michela Bonafede, Bruna M. Rondinone, Seoyeon Ahn, Emiko Ando, Jose Luis Ayuso-Mateos, Maria Cabello, Alexis Descatha, Nico Dragano, Quentin Durand-Moreau, Hisashi Eguchi, Junling Gao, Lode Godderis, Jaeyoung Kim, Jian Li, Ida E.H. Madsen, Daniela V. Pachito, Grace Sembajwe, Johannes Siegrist, Kanami Tsuno, Yuka Ujita, JianLi Wang, Amy Zadow, Sergio Iavicoli, Frank Pega, The effect of exposure to long working hours on depression: A systematic review and meta-analysis from the WHO/ILO Joint Estimates of the Work-related Burden of Disease and Injury, Environment International, Volume 155, 2021, 106629, ISSN 0160-4120, https://doi.org/10.1016/j.envint.2021.106629.

<Response>

Thank you for your valuable information. We have revised the literature on long working hours based on the references you provided (references No. 4–6).

-Job type category are not completely clear. What “Mainly labor” includes?  What “interpersonal communication” refers to? I suggest to explain better in the measures.

<Response>

The questionnaire includes examples of each job type, and the participants should have selected one that is as close as possible to their own job type. The examples are listed in the Materials and Methods section (page 4, lines 165–168). However, the classification of job type was not precise and it may have caused misclassification of job type categorization. We noted the possibility of misclassification as a limitation in the Discussion section (page 10, lines 323–329).

- At page 7 paragraph 2, authors stated: “Thus, workplaces should not only set criteria but also promote the reduction of long working hours during the COVID-19 pandemic”.

I think that promote the reduction of long working hours is not just a matter of during the Covid-19 pandemic.

<Response>

As you pointed out, reduction of long working hours is a necessary measure not only during the COVID-19 pandemic but also under normal situations. We have revised the relevant section to make this clear (page 9, lines 282–283).

- Authors include working from home as control variable and this is very appreciate in consideration of the topic. Nevertheless, there are several recent studies showing that working from home is increasing hours of work and reducing the sick leave absences at the same time (workers tend to work sick from home.). Maybe you should investigate (even only reporting frequencies) if the more the hour of working at home, the higher the long working hours are in your sample.

<Response>

We confirmed the association between the frequency of remote working and overtime work hours. As noted in Supplemental Table, the highest frequency of remote working (i.e., ≥ 4 times/week) people had the highest rate of answering almost no overtime work, while the almost low frequency remote working people (i.e., ≥ 1 time/month) had the highest rate of working ≥ 2hours overtime per day. As a clear tendency for remote working to increase overtime work hours was not observed, the association between the frequency of remote working and working hours were not reflected in the text this time. We appreciate your kind consideration.

Supplemental Table Association between the frequency of remote working and overtime work hours

Overtime work hours

Almost none

< 2 hours

≥ 2 hours

(n = 12,843)

(n = 11,740)

(n = 2,453)

n (%)

n (%)

n (%)

Frequency of remote working

  ≥ 4 times/week

 1,763 (63.2)

  797 (28.6)

 230 (8.2)

  ≥ 2 times/week

  616 (41.7)

  696 (47.1)

  165 (11.2)

  ≥ 1 time/week

  348 (35.9)

  437 (49.8)

   93 (10.6)

  ≥ 1 time/month

  221 (35.9)

  323 (52.5)

   71 (11.5)

  Almost none

9,895 (46.5)

9,487 (44.6)

1,894 (8.9)

We hope that the revised manuscript in now acceptable for publication in the International Journal of Environmental Research and Public Health. Thank you.

Round 2

Reviewer 3 Report

The authors have greatly improved the quality of the manuscript, modifying and adding to the considerations I made last time. I congratulate them on their work.